# What does the American public know about child marriage?

**David W. Lawson** *, Rachel Lynes, Addison Morris, Susan B. Schaffnit

Department of Anthropology, University of California, Santa Barbara, California, United States of America

* dlawson@anth.ucsb.edu

## Abstract

Global efforts to eradicate 'child marriage' (<18 years) increasingly target governments, the private sector and the general public as agents of change. However, understanding of child marriage may be subject to popular misconceptions, particularly because of ambiguity in the age threshold implied by the term 'child', and because awareness campaigns routinely emphasize extreme scenarios of very young girls forcibly married to much older men. Here, we ascertain public knowledge of child marriage via an online survey. Half of those surveyed mistakenly believed that the cut-off for child marriage is younger than the threshold of 18 years, and nearly three-quarters incorrectly believed that most child marriages occur at 15 years or below (it primarily occurs in later adolescence). Most participants also incorrectly believed that child marriage is illegal throughout the USA (it's illegal in only 4/50 states), substantially overestimated its global prevalence, and mistakenly believed that it primarily takes place among Muslim-majority world regions. Our results highlight important popular misconceptions of child marriage that may ultimately undermine global health goals and perpetuate harmful stereotypes. Organizations seeking to empower women by reducing child marriage should be cautious of these misunderstandings, and wary of the potential for their own activities to seed misinformation.

## Introduction

Rosling et al.'s 2018 book Factfulness [1], exposed widespread ignorance among not only the public, but also global health professionals, about the state of world we live in. In a simple multiple-choice questionnaire distributed to nearly 12,000 people in 14 countries, concerning topics such as global life expectancy, child vaccination rates, and gender parity in education, only 10% performed better than if they had chosen their responses at random [1]. The large majority of errors were caused by participants believing the state of the world was considerably worse than it actually is. Rosling et al. [1] attribute this pattern to the way that we receive and process information, most importantly the tendency for negative news to get media interest and retain our attention. Negative news may be effective at garnering public interest, but resulting biases in understanding can be counterproductive to global health objectives [1]. Public and policy-maker misconceptions can stifle effective decision-making, ultimately misprioritizing global health issues. Moreover, misrepresentations of low-income nations can

**Data Availability Statement:** All relevant data are within the paper and its Supporting Information files.

**Funding:** This study was funded by a National Science Foundation grant to DWL (Award Number:

1851317) and by the University of California, Santa Barbara. The funders played no role in the study design, data collection and analysis, decision to publish, or preparation of the manuscript.

**Competing interests:** The authors have declared that no competing interests exist.

reinforce inaccurate and potentially harmful stereotypes, which in turn are detrimental to sociopolitical relationships with wealthier 'donor' countries, including stances on migration, trade and violent conflict [1, 2]. Inspired by Rosling et al. [1], here we consider public understanding of one specific issue: child marriage, a topic of increasing prominence in global health.

Within international development, 'child marriage' refers to marriage under age 18 years. It most commonly affects girls, and is most prevalent in sub-Saharan Africa and South Asia, where an estimated 38% and 30% of girls marry before 18 years respectively [3]. Policy attention on child marriage has escalated in the past decade [4]. 'Girls Not Brides', for example, founded in 2011, represents a global partnership of now over 1,000 civil society organizations committed to ending child marriage and increasing public awareness of its prevalence and purported harmful consequences [5]. The United Nation's (UN) 2015 Sustainable Development Goals pledged to abolish child marriage within a generation, with an ambitious 15-year target [6]. Furthermore, worldwide Google searches for child marriage approximately doubled over the last 10 years, illustrating parallel escalating interest in child marriage in the general population (Fig 1).

Even with rising interest, we expect widespread misconceptions about child marriage among residents of the USA, or other high-income populations, for several reasons. Importantly, these misconceptions are predicted to be more than simple errors reflecting a lack of knowledge, but rather indicative of a biased understanding of the topic. First, well-intentioned public awareness campaigns may inadvertently misrepresent the realities of child marriage by presenting extreme scenarios most capable of capturing donor attention. Most notably, child brides are routinely portrayed as extremely young, often prepubescent girls, forcibly married to considerably older men. A systematic review of campaign imagery is beyond the scope of this study, but numerous illustrative examples exist in the public domain. For example, in UNICEF's 2019 'A story book proposal' video campaign, a middle-

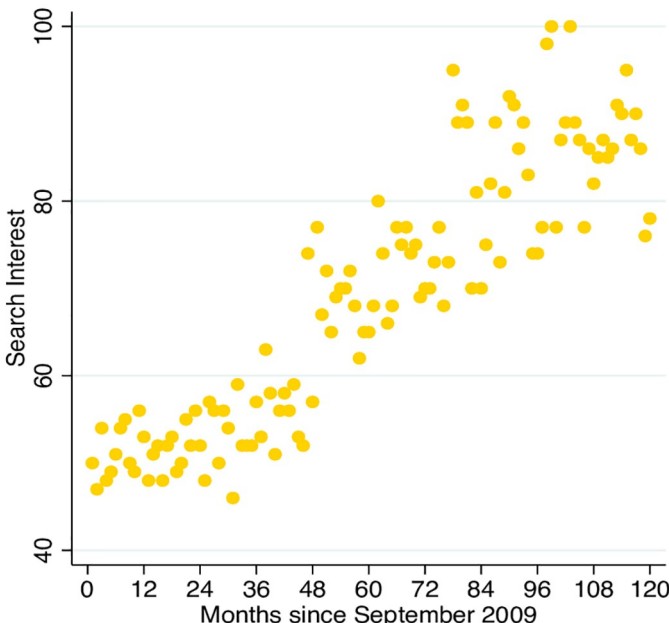

**Fig 1. Trends in worldwide Google searches for child marriage.** 'Search interest' measures the number of worldwide google searches for child marriage relative to the highest volume of searches (represented as 100) observed between September 2009 and 2019. Data are publicly accessible from trends.google.com.

aged man speaking directly to camera shares his excitement about his upcoming proposal [7]. The scene then shifts to a public location, where the man gives a payment to an adult woman before dragging away a distressed very young, ostensibly pre-pubescent girl. Statistics on the global prevalence of child marriage then appear on screen. Plan International's 2014 'Stop the Wedding' campaign, staged a mock church wedding of a 12-year girl to a 37-year-old man, with the event widely covered by the international media. This high-profile campaign won numerous awards for its attention-grabbing content [8]. A video of the ceremony, ending with the message "every day, 39,000 girls are *forced* into marriage", has been viewed by over 13 million times [9].

In contrast to these narratives, in every world region most 'child brides' are typically married in later adolescence, and rarely are spousal age gaps quite so wide. According to recent UNICEF statistics, worldwide 76% of marriages under 18 years take place at or over 15 years [3]. We therefore expect the emphasis on relatively extreme scenarios in the campaign materials described above will lead people to overestimate the prevalence of child marriage in line with the tendencies described by Rosling et al. [1], but moreover to specifically overestimate the prevalence of child marriages at very young ages. We also anticipate that the routine and explicit portrayal of young girls being coerced into marriage by men and/or their parents, will lead the public to believe that child marriages are inherently forced marriages. In contrast, research within communities where early marriage is commonplace demonstrates variability in girls' agency in the marital process, and that marriage under 18 years does not always imply arranged marriage, or coercion by parents or the husband [e.g. 10–14]. As such, while consent to marry at young ages may not always be best considered as 'informed consent', the portrayal of child marriages as inherently forced is questionable.

The second and related reason is terminology. The term child marriage is rooted in a human rights framework, taking the legal definition of childhood as ending at 18 years [15, 16]. Yet in popular discourse, and more generally across the social and health sciences, the term childhood is used inconsistently, but generally refers to the period before puberty, after which children become adolescents or teenagers, and then adults. The Collins English Dictionary [17], for example, defines childhood as "the condition of being a child; the period of life before puberty". As such, the general public may understandably be under the false assumption that child marriage refers specifically to the marriage of especially young, or pre-pubescent girls. Confusion may be further compounded by the inconsistent usage of terminology across behavioral domains. For example, pregnancy at similarly young ages is generally referred to as 'teen pregnancy' or 'teen motherhood' [18, 19].

Finally, child marriage is categorized as a 'harmful cultural practice', akin to female genital cutting/mutilation, presenting a moral obligation for intervention [20]. This framing invites donors to be outraged that girls/young women are forced into marriage to draw support for the end child marriage movement, implicitly positioning high-income countries as morally superior [21]. Coupled with broader ethnocentric misconceptions about low-income countries, we therefore anticipate the public will be surprised to discover that, despite escalating pressure placed on lower-income populations to ban child marriage, it remains legal in most high-income nations. At the time this study was conducted (early 2019), in the USA only New Jersey and Delaware had made marriage under the age of 18 years illegal, while it remained legal with parental consent and judicial approval in 48 states [22]. Today marriage under 18 years is legal in 46 out of 50 states [23].

To establish public knowledge of child marriage and ascertain directions of bias we implemented an internet survey conducted on Amazon Mechanical Turk (MTurk), an online tool used to recruit members of the general population. From the outset, we note that this is unlikely to be an entirely representative sample. MTurk workers are more educated, less

religious, and more likely to be unemployed than the general population [24]. Our results are therefore interpreted with appropriate caution. These issues aside, the reliability of data collected from MTurk has also been found to be not significantly different from data collected by traditional survey methodologies [25]. Moreover, MTurk offers a cost-effective data collection tool in the absence of pre-existing sources of data.

## Materials and methods

As an anonymous internet survey involving low risk to participants, the study was deemed exempt from ethical review by the University of California Office of Research. Consent was obtained by all participants completing the survey. Our survey recruited only USA residents of at least 18 years of age and who had a MTurk approval rating over 85%. Participants were paid $0.50 and the survey took approximately 10 minutes to complete. A target sample size of 750 individuals was selected to balance budgetary constraints with the benefits of a large sample, anticipating some degree of non-completion.

Participants were asked 10 questions pertaining to child marriage (see Results) and were instructed to respond based on their current knowledge, rather than use the internet or other sources before answering. Participants also provided sociodemographic characteristics (state of residence, sex, age, education level, employment status, and political leaning) and answered two test questions to confirm that they were giving considered responses. The first asked them to describe what they ate yesterday in a grammatically correct sentence. The second asked them to list three given colors in reverse order. Failure to answer these questions correctly led to the participant being excluded from analysis. Following the survey participants were provided accurate statistics on child marriage.

We present descriptive statistics for each of the 10 questions on child marriage alongside correct responses where available (primarily based on global statistics compiled by UNICEF [3]. We have no hypotheses regarding relationships between participant sociodemographic characteristics and their knowledge of child marriage. However, an exploratory analysis of potential differences offers a window into the scope for non-representative sampling to bias our results. Therefore, we also present supplementary analyses of the bivariate relationships between these variables, using ANOVA or chi-squared tests as appropriate.

## Results

Our survey was completed by 755 people. Excluding individuals who did not complete the full survey (n = 40) or provided invalid responses to either test question (n = 106), left 609 valid cases (Table 1, S1 Dataset). More than half (59%) identified as female, and ages ranged from 18 to 74 years (median = 34 years; interquartile range (IQR) = 27–45). A majority were university educated (55% had a bachelor's degree or higher), currently employed (79%) and born in the USA (95%). Participants included residents of 46 out of 50 states, and Washington DC. Political leanings were diverse but centrist on average; asking participants to place themselves on a scale ranging zero for completely liberal to 100 for completely conservative, the median score was 47 (IQR = 14–63).

Table 2 shows participant responses to child marriage questions alongside correct answers where they exist. Participants were asked to identify, to the best of their ability, the UN threshold for child marriage, and what age they believed childhood ends, with the option to answer any age in years. Fig 2A shows the distribution of responses for both of these questions (excluding 3 improbable answers for age marking the end of childhood). The median response for the UN threshold for child marriage was 17 years (IQR = 16–18). Thus, half (49%) of our participants correctly identified 18 years as the UN threshold defining child marriage, while

**Table 1. Characteristics of survey respondents (N = 609).**

| | median (IQR) / n (%)* |
|---|---|
| Sex | |
| Female | 359 (59%) |
| Male | 247 (41%) |
| Other | 3 (0%) |
| Age (years) | 37 (27–45) |
| Highest level of education | |
| Some high school | 5 (1%) |
| High school diploma/G.E.D. | 57 (9%) |
| Some college | 146 (24%) |
| Associate's degree | 66 (11%) |
| Bachelor's degree | 264 (43%) |
| Master's degree | 66 (11%) |
| Doctoral degree | 5 (1%) |
| Employment status | |
| Unemployed | 76 (12%) |
| Employed | 479 (79%) |
| Student | 32 (5%) |
| Retired | 22 (4%) |
| Political leaning (0 to 100, liberal to conservative) | 47 (14–63) |

* Number and percentage of cases are reported for categorical variables (Sex, Highest level of education and Employment status). Medians and interquartile range are reported for continuous variables (Age and Political leaning).

the other half (50%) guessed a younger threshold. Around one third (32%) answered that only marriages under the age of 16 years qualify as child marriages. Fewer participants suggested that childhood ends at the 18-year threshold (39%), with 47% of participant suggesting that childhood ends at or below 16 years.

Participants were then asked in what age ranges and in which regions of the world do most child marriages occur? Around two thirds (74%) of participants believed that child marriage was most common under 16 years, with 44% selecting ages 13–15 years. In reality, child marriages are most frequent just below 18 years of age, with 76% of child marriages taking place at or over 15 years of age [3]. Most participants (51%) incorrectly answered that child marriage is most common in the Middle East and North Africa. The next most common guess was the correct answer, with 15% of participants correctly identifying that most child marriages take place in sub-Saharan Africa.

Participants were asked in how many US states is child marriage currently legal. Just under half (42.5%) thought that child marriage was illegal in all states, and almost all respondents guessed it was legal in 5 or fewer states (IQR = 0–8) (Fig 2B). At the time of survey, the marriage of minors was only banned in two states [22]. Participants were asked what proportion of women marry before 18 years of age in sub-Saharan Africa, South Asia (the two world regions with the highest rates of child marriage), and in the USA today and in the 1950s. Participants overestimated its frequency in Sub-Saharan Africa and South Asia (Fig 2C). They also substantially overestimated its prevalence in the USA today and in the past (Fig 2D).

Finally, participants estimated how many child marriages they believed to be forced. The large majority felt that child marriage was forced most or all of the time. We are aware of no

**Table 2. Responses of survey participants (n = 609) and correct answers.**

| Survey Question | Responses [median (IQR) / n (%)]* | Correct Answer |
|---|---|---|
| **Q1**. What is the legal threshold for 'child marriage' as defined by the UN? | 17 (16, 18) | 18 years [3] |
| **Q2**. What age marks the end of childhood? | 17 (14, 18) | Legally, 18-years [15], but the terms 'child' and 'childhood' are used inconsistently throughout the social and health sciences. |
| **Q3**. In which age range do most 'child marriage' occur? | | Child marriages typically occur most frequently in late adolescence, just under the threshold of 18 years. According to the most recent UNICEF statistics three quarters (76%) of child marriages globally occur at or above 15 years [3]. |
| Under 10 years | 36 (6%) | |
| 10–12 years | 146 (24%) | |
| 13–15 years | 266 (44%) | |
| 16–18 years | 135 (22%) | |
| 19–21 years | 26 (4%) | |
| **Q4**. Child marriage is legal in how many of the 50 US states? | 2 (0, 8) | At the time of survey, the marriage of minors was only completely banned in two states. In the remaining 48 states, marriage under 18-years was allowed in various circumstances [22]. |
| **Q5**. In which world region is 'child marriage' most common? | | Child marriage is most common sub-Saharan Africa & South Asia [3]. |
| Central Asia | 37 (6%) | |
| East Asia and Pacific | 64 (11%) | |
| Europe | 8 (1%) | |
| Middle East and North Africa | 313 (51%) | |
| North America | 34 (6%) | |
| South Asia | 63 (10%) | |
| Sub-Saharan Africa | 90 (15%) | |
| **Q6**. What proportion of women marry before age 18 years in sub-Saharan Africa? | 48 (30, 67) | The most recent estimates indicate that 38% of women marry before age 18 in sub-Saharan Africa [3]. |
| **Q7**. What proportion of women marry before age 18 years in South Asia? | 40 (24, 55) | The most recent estimates indicate that 30% of women marry before age 18 in South Asia [3]. |
| **Q8**. What proportion of women marry before age 18 years in the US? | 10 (5, 20) | Approximately 0.6% of American women are married under 18-years [26]. |
| **Q9**. What proportion of women married before age 18 years in the US in the 1950s? | 30 (17, 44) | Approximately 15–20% of American women married under age 18-years throughout the 1950s [27]. |
| **Q10**. How often are child marriages forced? | | According to some legal frameworks all marriages under age 18 years are forced because minors are understood to lack agency and/or informed choice. However, ethnographic work from around the world demonstrates varying levels of agency among girls entering marriages ranging from none to a great amount (see main text). |
| Always | 183 (30%) | |
| Most of the time | 378 (62%) | |
| Rarely | 46 (8%) | |
| Never | 2 (0.3%) | |

* Medians and interquartile range are reported for continuous questions (Q1, Q2, Q4, Q6, Q7, Q8, Q9). Number and percentage of cases are reported for categorical questions (Q3, Q5, Q10).

data which speaks to what percentage of marriages (of minors or adults) worldwide are forced (see Discussion).

Bivariate associations between participant characteristics and responses reveal a number of relationships (S1 to S5 Tables in S1 File). Women, older, currently unemployed and retired

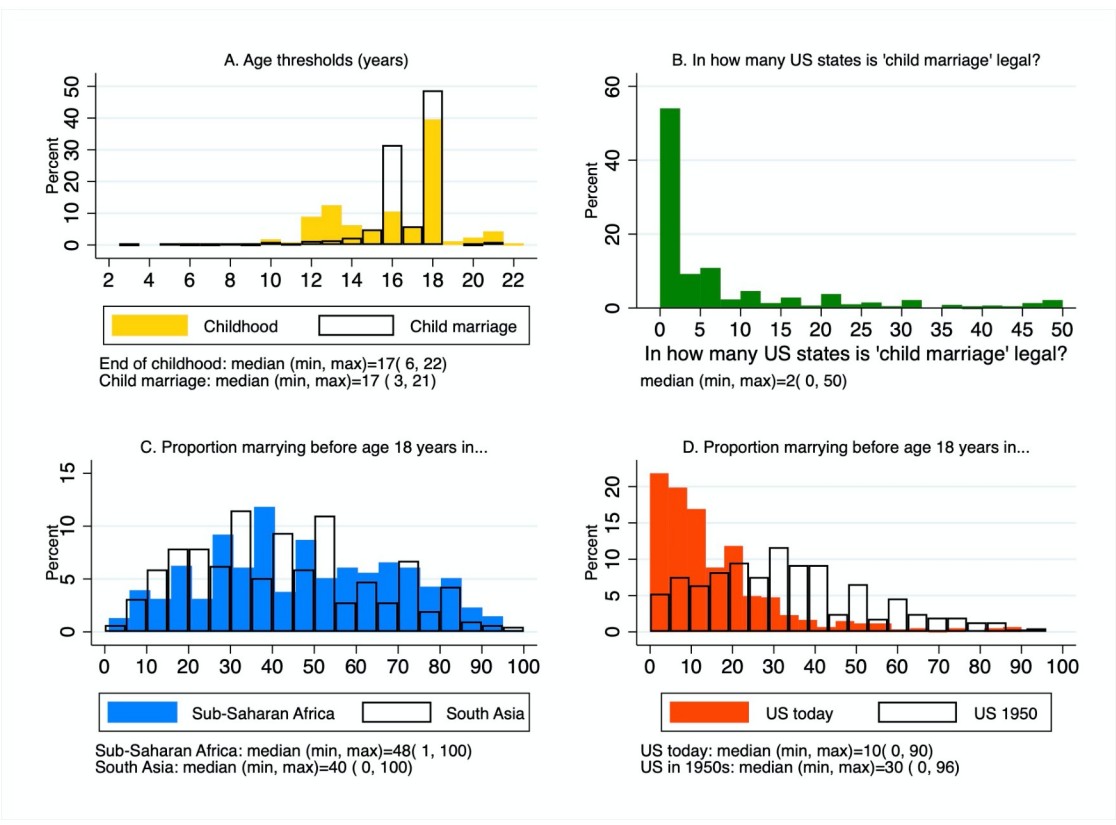

**Fig 2. Histograms showing distribution of participant responses. A)** responses to the questions: "Q1. What is the legal threshold for 'child marriage' as defined by the UN?" and "Q2. What age marks the end of childhood?". **B)** responses to the question: "Q4. Child marriage is legal in how many of the 50 US states?". **C)** responses to the questions: "Q6. What proportion of women marry before age 18 years in sub-Saharan Africa?" and "Q7. What proportion of women marry before age 18 years in South Asia?"; **D)** responses to the questions: "Q8. What proportion of women marry before age 18 years in the US?" and "Q9. What proportion of women married before age 18 years in the US in the 1950s?"

participants showed a greater tendency to overestimate the proportion of women married before age 18-years in sub-Saharan Africa and South Asia compared to men, younger participants and those who are currently students or employed (respectively). Women were also more likely to say that child marriages are forced always or most of the time, and more likely to guess that child marriages mostly take place at ages under 12 years than men. Participants with a higher education guessed that child marriage was legal in more US states than those with lower education, though they still vastly underestimated the correct answer. More educated respondents were less likely to misidentify the Middle East and North Africa as the regions with the highest rate of child marriages.

## Discussion

Our results suggest that the American public are poorly informed about child marriage. This in itself is not surprising; public understanding of the state of the world is generally low and our observations are in line with the widespread ignorance about global health issues demonstrated by Rosling et al. [1]. What is interesting, however, is the direction of common errors, revealing biases in popular understanding. These misconceptions have important implications for initiatives addressing child marriage and aiming to empower girls and young women in both relatively low and high-income settings. Most strikingly, while close to half answered

correctly, the majority of participants believed that the cut-off for child marriage is younger than the legal threshold of 18 years, and nearly three-quarters incorrectly believed that most child marriages occur at 15 years or below, despite the fact that child marriage primarily takes place in later adolescence worldwide [3]. With many participants also answering that they believe childhood ends earlier than 18 years, it seems likely that the specific terminology of child marriage contributes to this confusion.

Notably, the term child marriage is also used with reference to early marriages in high-income populations [e.g. 26, 28], but not universally so. Indeed, while terminology appear to be becoming more consistently used as the end child marriage movement has accelerated, marriage at equivalent ages in the USA and Europe, has often been referred to as 'teen marriage' (ususally referencing an alternative threshold of <20 years, [e.g. 27, 29], 'early marriage' [e.g. 30, 31] or 'adolescent marriage' [e.g. 32, 33]. Such terminology conjures notions of relatively older girls/young women with a higher degree of autonomy in the decision to marry. The reasons behind, and implications of, this discrepancy in terminology require consideration.

A comparison of child marriage to teen pregnancy is instructive. Pregnancy under the age of 20 years is labelled 'teen pregnancy' in global health discourse across low and high-income contexts. As such, a pregnant 17-year-old is labelled a 'teen', but if married, and especially if she is from a low-income country, she is a labelled 'child'. This distinction is likely because interest in teen pregnancy initiated in reference to sexual and reproductive health in high-income societies, before later being extended into global health scholarship [34, 35]. In contrast, interest in child marriage gained momentum first through concerns about female wellbeing in relatively poorer nations, originating out of an international human rights framework approaching the notion of a 'child' as a legal entity [4, 16]. The adoption of a legal 18-year threshold between childhood and adulthood was a logical consequence of a series of changes relating to childhood in high-income countries which had come about over nearly a century, such as the implementation of child labor laws and mandatory schooling [27]. In contrast, while the same threshold is utilized by lower-income nations who now share global targets addressing early marriage [36], 18 years is an arguably arbitrary cut-off in contexts without these shared historical legal changes surrounding concepts of childhood.

What implications might there be of the public overestimating the prevalence of very early child marriages? On the one hand, it might have the desirable outcome of creating a sense of urgency and increasing commitments to the global health target to end child marriage. But this benefit comes at the potential cost of perpetuating harmful stereotypes. In particular, it reinforces views of girls/young women in low-income countries as passive victims rather than active agents in their own lives. Since child marriages are mostly assumed to occur via exploitative coercion by parents and husbands, overestimating the prevalence of child marriage may also reinforce wider views of low-income nations as in need of moral rescue from their own cultural traditions. Such representations can counterintuitively undermine humanitarian empathy and encourage ethnocentric judgements of low-income countries as to blame for their own hardships [2], and more generally stifle consideration of the broader structural factors (e.g. poverty, lack of viable alternatives) detrimental to girls and women [37, 38, see also 39]. Future 'end child marriage' campaigns would do well to weigh the benefits of emphasizing extreme cases to garner research, philanthropic and policy attention against the less tangible dangers of promoting stereotypes that will not resonate in all communities and may generate misunderstanding.

Whether all marriages under 18 years should best be considered 'forced marriages' is somewhat controversial, and we not aware of data capable of categorizing marriages globally as 'forced', 'arranged', or 'free'. As such, we cannot evaluate whether or not participants are

correct to view most child marriages as forced. Nevertheless, responses to this particular question suggest that a significant proportion of the general public would be surprised to learn that female adolescents in multiple regional and cultural contexts marry without clear coercion from parents or husbands, and in some cases, such as in cases of elopement, even against the wishes of their family [10–13, 40–42].

The large majority of participants also believed that child marriage is illegal throughout the majority of the USA. While direct interpretation of this result is complicated by the fact that many participants misidentified the age threshold for child marriage (Fig 2), this result suggests widespread ignorance that marriage before 18 was illegal in only two states at the time of survey. This has at least two implications. First, organizations such as Unchained at Last [43] which campaign to criminalize American child marriage, should act on the assumption that the public are largely unaware that it remains legal across almost the whole country. Second, it is evident that many Americans are likely unaware of the hypocrisy inherent to their county's position on the topic. The USA and other high-income nations put considerable pressure on low-income nations to adopt and enforce laws regarding child marriage. Yet, recent efforts to raise legal ages at marriage in the USA have been largely unsuccessful partly due to the sentiment that pregnant teenagers should have the option of marriage [44]. Such contingencies are notably absent from the USA's exported position on marriage under 18 years.

Survey respondents overestimated the prevalence of child marriage abroad and at home, consistent with a general tendency to overestimate the prevalence of undesirable outcomes [1]. These misconceptions could result from outdated viewpoints rather than outright errors, since the prevalence of child marriage has declined globally. However, public interest in child marriage has risen dramatically in recent years (Fig 1), so that we might expect intuitions to be based on recent statistics, rather than outdated ones. Participants also had a poor sense of the regional distribution of child marriage worldwide. Age at marriage trends vary within the crude world region categories considered in this survey [see 45], which may have undermined public knowledge, but it is remarkable that the majority incorrectly answered that child marriage primarily takes place among Muslim-majority regions of the Middle East and North Africa. This misconception could reflect wider stereotypes about Islam and the position of women within Muslim culture. Though speculative, this interpretation is consistent with patterns of Islamophobia documented in the USA [46].

An important limitation of this study is the use of MTurk. The fact that participant characteristics such as education level were related to their responses suggests that we should be cautious about extrapolating our results to the wider population. We used a modest threshold for MTurk respondent approval ratings (85%, see Materials and methods), which may have compromised data quality. Our methodology also cannot pinpoint the source of misconceptions, nor evaluate the potential role of end child marriage campaigns in seeding misinformation. It would be useful in future work to consider variation in public understanding in relation to such factors as experience of other cultures, educational/professional background in global health, and exposure to the global goal to end child marriage.

These reservations aside, action can be taken to limit public misconceptions. We advocate that the term child marriage is replaced with terminology which better synchronizes public understanding with reality. 'Adolescent', 'early' or 'minor marriage' are solid alternatives, with the latter alluding to the legal entity of the child upon which the age threshold is premised. Each term holds its own potential for misinterpretation, and we caution that there is no reason to prioritize public understanding among high-income nations specifically. Alternatively, child marriage could be used with reference only to marriage at especially young ages, such as under the age of 15 or 16 years. These considerations are also relevant to wider debates about the definitions of 'childhood' and 'adolescence'. For example, following accelerated physical

maturation paired with delayed social maturation in high-income countries it has recently been argued that the public health definition of 'adolescence' should be extended to refer to the ages of 10–24 years [47]. These discussions should include a consideration of popular (mis)understanding, and an acknowledgement of cultural variation in the timing of life transitions.

In conclusion, our results suggest that public understanding of child marriage is poor, but more importantly, shaped by wider misperceptions of both high and low-income nations. We therefore advocate for greater critical engagement with current terminology and dominant narratives within the end child marriage movement. Potential benefits of this engagement include greater awareness of the potentially distinct drivers and wellbeing implications of marriage in early versus later adolescence [48], and the reality that some girls and young women actively choose to marry early, even if this choice may be poorly informed and/or has negative consequences [12, 40]. As Bunting [48] argues, an inflexible strategy grounded in a rights-based approach to a uniform marriage age risks missing the complexity of *both marriage and age*. These considerations do not deny the existence or seriousness of forced and early marriages, or that coercion, agency, consent and the emergence of autonomy among young people are complicated phenomena to define and quantify [16]. They do, however, avoid the promotion of stereotypes which may, in some contexts, unfairly villainize parents and husbands, and stigmatize young people who choose to marry. Ultimately, this can only lead to more culturally-sensitive and more effective global health policy and practice.

## Supporting information

**S1 Dataset.**
(CSV)

**S1 File.**
(PDF)

## Acknowledgments

We thank Michael Barlev and Spencer Mermelstein for guidance on using MTurk, and Alissa Koski for constructive critique on an early version of this paper.

## Author Contributions

**Conceptualization:** David W. Lawson.

**Data curation:** Rachel Lynes, Addison Morris.

**Funding acquisition:** David W. Lawson.

**Investigation:** David W. Lawson, Rachel Lynes, Addison Morris, Susan B. Schaffnit.

**Methodology:** David W. Lawson, Susan B. Schaffnit.

**Project administration:** David W. Lawson, Susan B. Schaffnit.

**Supervision:** David W. Lawson, Susan B. Schaffnit.

**Writing – original draft:** David W. Lawson.

**Writing – review & editing:** David W. Lawson, Susan B. Schaffnit.

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
