## [Decision Letter · Decision Letter 0]

21 Jul 2020

PONE-D-20-17482

What does the American public know about ‘child marriage’?

PLOS ONE

Dear Dr. Lawson,

Thank you for submitting your manuscript to PLOS ONE. After careful consideration, we feel that it has merit but does not fully meet PLOS ONE’s publication criteria as it currently stands. Therefore, we invite you to submit a revised version of the manuscript that addresses the points raised during the review process.

Overall, the reviewers had very positive comments about this manuscript. They had some small concerns to be addressed....

We look forward to receiving your revised manuscript.

Kind regards,

Mellissa H Withers, PhD, MHS

Academic Editor

PLOS ONE

Journal Requirements:

Reviewers' comments:

Reviewer's Responses to Questions

**Comments to the Author**

1. Is the manuscript technically sound, and do the data support the conclusions?

Reviewer #1: Yes

Reviewer #2: Yes

Reviewer #3: Yes

2. Has the statistical analysis been performed appropriately and rigorously? 

Reviewer #1: Yes

Reviewer #2: Yes

Reviewer #3: Yes

3. Have the authors made all data underlying the findings in their manuscript fully available?

Reviewer #1: Yes

Reviewer #2: Yes

Reviewer #3: Yes

4. Is the manuscript presented in an intelligible fashion and written in standard English?

Reviewer #1: No

Reviewer #2: No

Reviewer #3: Yes

5. Review Comments to the Author

Reviewer #1: I fully enjoyed this ms, particularly as I live in a culturally mixed community where there are many marriages among people just under 18 years of age. These couples form tight bonds and begin families early. They are far from what I consider "child" marriages.

Despite this, there are some editorial issues that detract from the ms. Specifically:

1) Sentences should not begin with Arabic numerals. See any style manager such as https://style.mla.org/percentage-at-start-of-sentence/. Thus, please rewrite sentences beginning on lines 156, 158, 181, 194, and 202. I am uncertain that I caught all of them, but be sure to address this issue before resubmission.

2) Table titles should appear above the tables, not within them. They also need to be complete so that the table can be understood if presented on its own. Again, see any style manager such as APA which states: Each table and figure must be intelligible without reference to the text, so be sure to include an explanation of every abbreviation (except the standard statistical symbols and abbreviations). Thus, please remove the table titles from within the tables and restate them so they comport with table title requirements. For example, Table 1 - Participant characteristics for what? Table 2 - Participant responses to what? Check the table titles in several journals to see what I mean.

Table 1 Column 2 is mislabeled and confusing to the reader. Change n (%) and insert a row above Political Leanings and insert median (IQR).

Table 2 Column 2 is also confusing. Again, insert rows to indicate which are n (%) and which are median (IQR) or perhaps relabel the column Responses and indicate the format in the question such as:

Q1. What is the legal threshold for 'child marriage; as defined by the UN? [median(IQR)]

What is the format for Q9? It is listed as 30 (17. 44). Should this be a comma or closed space?

3) Lines 283 and 284 are redundant. Please remove the in-line citations.

4) Line 293 uses US rather than USA. Please be consistent. When referring to the proper noun use USA and reserve US for the adjective here and throughout the ms.

5) Line 305 should be e.g.

6) Line 402 - align your citation

References:

These are not standardized nor well proofed. All journal titles should be caps. All article titles should be sentence case. For example:

356 - remove space before :

361 - No journal listed

366 - Cap journal title

370 - remove space before .

375 - What does the 9. mean?

387 - Article not in sentence case

388 - Journal should not be in ital

389 - Why is this ital?

I stopped here. Please standardize and proof ALL your references.

Supplementary Tables

Should be (s.d.) throughout, not (s.d)

Also - somewhere I saw a8 rather than 18, but I cannot find it again.

Good luck with your resubmission!

Reviewer #2: This manuscript uses survey data to show how Americans profoundly misunderstand exactly what “child marriage” entails, and how common it is around the world. The results are interesting and important, and the manuscript will make a solid contribution after mostly cosmetic revisions.

Title: There’s no good reason child marriage should be in quotes. The scholarly proclivity for precision—these marriages usually involve teenagers, not children—is less important than just using commonly understood English language. The abstract indeed makes clear just what child marriage means in the world. Moreover, the quotes are repeated inconsistently throughout the text and the figures/tables: sometimes it’s child marriage, sometimes it’s “child marriage.”

p. 3, ln. 52. Negative news may garner interest, donations, and support for foreign aid. Or not. I don’t know the answer, so I don’t think the authors should presume to know it either.

p. 6, ln. 121. The last time I checked, minors could marry in some states under two other conditions, pregnancy and with judicial approval.

p. 7, ln. 135. Is it normal practice to only use Turkers with 85%+ approval ratings? How might this affect the results? Are there demographic attributes that distinguish highly rated Turkers? Similarly, did non-compliant Turkers give different answers on the child marriage questions (lns. 142-146)? Both of these seem like conventional concerns about selection bias that should be probed with the data at hand. I’m not asking for a properly identified instrumental variable model here, but just some bivariate statistics.

p. 7, ln. 135. I realize that a free market is a free market, but I find it troubling that the authors chose to compensate survey respondents at what translates to a wage of $3 an hour. If $325 is all they could afford, fine. Otherwise I’d urge them to pay their respondents a more ethical wage next time. Obviously this has no bearing on whether the manuscript should be published, but I’m perfectly OK with using my status as a reviewer to encourage the authors to do better in the future

p. 8, lns. 158, 160, 161, passim. Round off percentages to whole numbers. Decimals generally imply better accuracy in measurement than is generally possible. They also convey no useful information, but provide eye clutter. Exceptions to this rule should only be made when the decimals represent meaningful increments. Examples: the unemployment rate, or regression coefficients.

p. 8, lns. 162-164. Political leanings of 47 on a 0-100 scale are more meaningfully described as centrist than as “slightly left of center.”

Table 1 is far more bewildering than a table of summary statistics needs to be. Why in the world are IQRs presented? Just present means or medians for continuous variables and percentages for categorical variables in a format that doesn’t require at me to stare at the table for 20 seconds to figure out, say, what percent of the sample has a high school diploma.

p. 9, lns. 177-178. The option to answer any question in years??? How else are people going to answer? When was the last time you told anyone your age in months? This is nonsensical.

p. 9, lns. 179-182. If the correct answer is 18 and the median response is 17, then, yes, 50 percent of people provided answers of under 18. That is how medians work. In case it’s not apparent, my point is that there’s something about how the authors are presenting these results that defies logical exposition.

For Figure 2.A, there’s only a single threshold, not multiple thresholds. What’s more, tables and figures should always stand on their own: I should be able to look at one and make sense of it without consulting the text (and vice versa). That isn’t the case here, especially with Figures 2.C and 2.D. Is this the percentage of respondents who think each listed percentage in the figure is marrying before age 18? That’s a confusing question, because it’s not clear what the figures are representing.

p. 11, ln. 208. If people are overestimating both in the U.S. and in Africa and Asia, “but” is not the right conjunction.

p. 11, ln. 215. Older X unemployed X overestimating is three variables, so this isn’t bivariate analysis.

p. 12, lns. 226-228. The authors might observe that Americans are under-informed about just about everything: there are ample studies showing public ignorance across a wide range of topics.

p. 13. The authors are tying themselves in knots here to avoid saying the obvious: there are places on earth where nobody is too offended when two seventeen-year-olds get married. Similarly, most Americans might question the prudence and perhaps the decorum of an 18 year old American marrying a seventeen-year-old, but most people would probably hesitate to call it a “forced” marriage.

p. 15. In writing of hypocrisy, the authors are conflating individual belief and practice with the legal regime in the states they reside.

p. 15, ln. 296. The article cited [43] doesn’t establish that the right of pregnant teens to marry is the reason the laws haven’t changed; it’s just one possible reason.

p. 15, ln. 298. Say “respondent,” not “participant.” The latter is vague.

p. 16, lns. 322-323. The author’s faux anti-colonialist posturing here is fatuous. The authors are Americans writing for a predominantly Western audience. Foreign aid is coming from the West. Please omit this statement about whose understanding should be prioritized.

Reviewer #3: This manuscript addresses an important issue and is timely. However, at times, it verges on advocacy rather than scientific reporting. The authors might want to re-read the discussion with this in mind.

There is a missing word in line 248 "such, a pregnant 17-year-old is labelled a ‘teen’, but if married, and especially if she from a low." Please insert "is"after 'she."

6. PLOS authors have the option to publish the peer review history of their article (what does this mean?). If published, this will include your full peer review and any attached files.

Reviewer #1: No

Reviewer #2: No

Reviewer #3: No

---

## [Author Response · Author response to Decision Letter 0]

28 Jul 2020

Dear Editor,

Thank you for considering our manuscript. We are delighted to receive such positive and constructive commentary! Below we respond to each comment. Required changes are minor cosmetic issues, therefore we hope that you will now deem the manuscript publishable. 

Reviewer #1: I fully enjoyed this ms, particularly as I live in a culturally mixed community where there are many marriages among people just under 18 years of age. These couples form tight bonds and begin families early. They are far from what I consider "child" marriages. Despite this, there are some editorial issues that detract from the ms. Specifically:

1.1 Sentences should not begin with Arabic numerals. See any style manager such as https://style.mla.org/percentage-at-start-of-sentence/. Thus, please rewrite sentences beginning on lines 156, 158, 181, 194, and 202. I am uncertain that I caught all of them, but be sure to address this issue before resubmission.

Response: We have edited each of these sentences accordingly. Thank you! 

1.2 Table titles should appear above the tables, not within them. They also need to be complete so that the table can be understood if presented on its own. Again, see any style manager such as APA which states: Each table and figure must be intelligible without reference to the text, so be sure to include an explanation of every abbreviation (except the standard statistical symbols and abbreviations). Thus, please remove the table titles from within the tables and restate them so they comport with table title requirements. For example:

- Table 1 - Participant characteristics for what? 

- Table 2 - Participant responses to what? Check the table titles in several journals to see what I mean.

- Table 1 Column 2 is mislabeled and confusing to the reader. Change n (%) and insert a row above Political Leanings and insert median (IQR).

- Table 2 Column 2 is also confusing. Again, insert rows to indicate which are n (%) and which are median (IQR) or perhaps relabel the column Responses and indicate the format in the question such as:Q1. What is the legal threshold for 'child marriage; as defined by the UN? [median(IQR)]

- What is the format for Q9? It is listed as 30 (17. 44). Should this be a comma or closed space?

Response: We have edited the table titles as instructed. We have also added footnotes to clarify when we are quoting number and percentage of cases vs. medians and interquartile ranges. We have added a “%” symbol after percentages to bring further clarity to readers. We prefer this format to separating out categorical and continuous variables entirely into different tables or sub-tables as it allows us to retain the question order as they appear in the survey and in our results section – which ultimately improves the readability of the manuscript. Note - Q9 is a continuous response variable. We have replaced the period with a comma. Apologies for the error.

We have opted to retain table titles as the first row of each table – only so as to be clear to differentiate the table titles from the main text. Tables are ultimately formatting by the journal, so this should not be an issue. 

1.3 Lines 283 and 284 are redundant. Please remove the in-line citations.

Response: Corrected. 

1.4 Line 293 uses US rather than USA. Please be consistent. When referring to the proper noun use USA and reserve US for the adjective here and throughout the ms.

Response: Corrected. 

1.5 Line 305 should be e.g.

Response: Corrected 

1.6 Line 402 - align your citation

Response: Corrected. 

1.7 References: These are not standardized nor well proofed. All journal titles should be caps. All article titles should be sentence case. For example:

356 - remove space before; 361 - No journal listed; 366 - Cap journal title; 370 - remove space before .; 375 - What does the 9. mean?; 387 - Article not in sentence case; 388 - Journal should not be in ital; 389 - Why is this ital?; I stopped here. Please standardize and proof ALL your references.

Response: Apologies for this. We had some problem with our reference manager software. All details should now be complete. 

1.8 Supplementary Tables

Should be (s.d.) throughout, not (s.d)

Also - somewhere I saw a8 rather than 18, but I cannot find it again.

Response: Corrected. 

REVIEWER#2: This manuscript uses survey data to show how Americans profoundly misunderstand exactly what “child marriage” entails, and how common it is around the world. The results are interesting and important, and the manuscript will make a solid contribution after mostly cosmetic revisions.

2.1 Title: There’s no good reason child marriage should be in quotes. The scholarly proclivity for precision—these marriages usually involve teenagers, not children—is less important than just using commonly understood English language. The abstract indeed makes clear just what child marriage means in the world. Moreover, the quotes are repeated inconsistently throughout the text and the figures/tables: sometimes it’s child marriage, sometimes it’s “child marriage.”

Response: We have removed the quotes from the title and body of the text, with the exception that at retain them ONLY at the first definition of child marriage in the abstract and introduction. 

2.2 p. 3, ln. 52. Negative news may garner interest, donations, and support for foreign aid. Or not. I don’t know the answer, so I don’t think the authors should presume to know it either.

Response: We have changed “is” to “may be” for clarification, and included a citation to Rosling et al who provides substantial evidence for this point. 

2.3 p. 6, ln. 121. The last time I checked, minors could marry in some states under two other conditions, pregnancy and with judicial approval.

Response: We have now specified with judicial approval in the text. 

2.4 p. 7, ln. 135. Is it normal practice to only use Turkers with 85%+ approval ratings? How might this affect the results? Are there demographic attributes that distinguish highly rated Turkers? Similarly, did non-compliant Turkers give different answers on the child marriage questions (lns. 142-146)? Both of these seem like conventional concerns about selection bias that should be probed with the data at hand. I’m not asking for a properly identified instrumental variable model here, but just some bivariate statistics.

Response: We are not sure what counts as normal practice with use of MTurk, but believe 85% approval ratings to be acceptable. Nevertheless, we have now included the following sentence in discussion of limitations – “We used a modest threshold for MTurk respondent approval ratings (85%, see Materials and Methods), which may have compromised data quality.” Non-complaint MTurk respondents (i.e. those that failed the test questions) were not included in any of our analysis and so do not affect our conclusions. Most likely their data is of lower quality (as has been demonstrated in prior studies), but it is unclear to us what an analysis of their responses could add to the manuscript. It is standard practice to just exclude such cases from analysis, as we have done here. 

2.5 p. 7, ln. 135. I realize that a free market is a free market, but I find it troubling that the authors chose to compensate survey respondents at what translates to a wage of $3 an hour. If $325 is all they could afford, fine. Otherwise I’d urge them to pay their respondents a more ethical wage next time. Obviously this has no bearing on whether the manuscript should be published, but I’m perfectly OK with using my status as a reviewer to encourage the authors to do better in the future

Response: Thank you for bringing this to our attention. To be entirely honest this was our first foray into MTurk sampling and we simply followed the recommendations of colleagues in psychology that rely on this service. We now realize our rate of pay was unethical and are committed to doing better in future research! 

2.6 p. 8, lns. 158, 160, 161, passim. Round off percentages to whole numbers. Decimals generally imply better accuracy in measurement than is generally possible. They also convey no useful information, but provide eye clutter. Exceptions to this rule should only be made when the decimals represent meaningful increments. Examples: the unemployment rate, or regression coefficients.

Response: We have removed decimal places throughout. 

2.7 p. 8, lns. 162-164. Political leanings of 47 on a 0-100 scale are more meaningfully described as centrist than as “slightly left of center.”

Response: Corrected. 

2.8 Table 1 is far more bewildering than a table of summary statistics needs to be. Why in the world are IQRs presented? Just present means or medians for continuous variables and percentages for categorical variables in a format that doesn’t require at me to stare at the table for 20 seconds to figure out, say, what percent of the sample has a high school diploma.

Response: Interquartile ranges are appropriate and useful statistics in the interpretation of skewed data. This is not controversial. So we prefer to include them here. We have clarified which rows display medians and interquartile ranges, and which rows show number and percentage of cases (see also reply to point 1.2)

2.9 p. 9, lns. 177-178. The option to answer any question in years??? How else are people going to answer? When was the last time you told anyone your age in months? This is nonsensical.

Response: At worst this is a case of providing too more detail – we disagree that is nonsensical to provide this information and prefer to include keep the sentence as it is for purposes of clarity – it tells the reader the level of precision offered in our survey methodology. 

2.10 p. 9, lns. 179-182. If the correct answer is 18 and the median response is 17, then, yes, 50 percent of people provided answers of under 18. That is how medians work. In case it’s not apparent, my point is that there’s something about how the authors are presenting these results that defies logical exposition.

Response: We have edited the offending sentences for clarity. 

2.11 For Figure 2.A, there’s only a single threshold, not multiple thresholds. What’s more, tables and figures should always stand on their own: I should be able to look at one and make sense of it without consulting the text (and vice versa). That isn’t the case here, especially with Figures 2.C and 2.D. Is this the percentage of respondents who think each listed percentage in the figure is marrying before age 18? That’s a confusing question, because it’s not clear what the figures are representing.

Response: We have now totally revised the figure legend for Figure 2(A-D) – now quoting this specific question asked in full. We feel that with these changes everything should now be clear! 

2.12 p. 11, ln. 208. If people are overestimating both in the U.S. and in Africa and Asia, “but” is not the right conjunction.

Response: Corrected. 

2.13 p. 11, ln. 215. Older X unemployed X overestimating is three variables, so this isn’t bivariate analysis.

Response: These ARE bivariate associations. We do not examine interactions between employment and age. We examine each variable’s independent association with the responses. An absent comma seems to be responsible for this confusion – we have edited the text to clarify. 

2.14 p. 12, lns. 226-228. The authors might observe that Americans are under-informed about just about everything: there are ample studies showing public ignorance across a wide range of topics.

Response: We have added this observation and our existing citation to Rosling et al. covers this issue extensively. 

2.15 p. 13. The authors are tying themselves in knots here to avoid saying the obvious: there are places on earth where nobody is too offended when two seventeen-year-olds get married. Similarly, most Americans might question the prudence and perhaps the decorum of an 18 year old American marrying a seventeen-year-old, but most people would probably hesitate to call it a “forced” marriage.

Response: There is no clear recommendation here. We emphasize that while our observations might seem obvious to the enlightened this does not change the fact that current efforts to end child marriage rarely exchange with this reality. As such, exposition of the issues at hand is warranted and fully justified here. 

2.16 p. 15. In writing of hypocrisy, the authors are conflating individual belief and practice with the legal regime in the states they reside.

Response: We fail to see the point here. Hypocrisy can apply to actions not just individual beliefs and the American public are complicit in their own legal regime and support for the end child marriage campaign applied to other countries. It is after all widespread support for the right to marry under 18 years if pregnant that has led to the retention of laws that allow child marriage within the USA. See the in text reference to the article: “Clark 2018 End child marriage in the U.S.? You might be surprised at who’s opposed”. We could add a discussion of individual belief vs national and state laws, but this seems unnecessary.

2.17 p. 15, ln. 296. The article cited [43] doesn’t establish that the right of pregnant teens to marry is the reason the laws haven’t changed; it’s just one possible reason.

Response: We have clarified this by adding “partly”. 

2.18 p. 15, ln. 298. Say “respondent,” not “participant.” The latter is vague.

Response: Changed. 

2.19 p. 16, lns. 322-323. The author’s faux anti-colonialist posturing here is fatuous. The authors are Americans writing for a predominantly Western audience. Foreign aid is coming from the West. Please omit this statement about whose understanding should be prioritized.

Response: There is nothing faux about it. The point is important – we are arguing that paying attention to (a)synchrony between public understanding of terminology and its use by policy-makers and development organizations is important if we want to minimize the potential harms of such misunderstandings (as highlighted in our opening paragraph). Since efforts to end child marriage primarily relate to low and middle-income countries it is worth emphasizing that understanding among their public is important, rather than simply the American public. Please also note a reviewer on a previous version of this manuscript actually asked us to explicitly include this point so as to avoid centralizing a Western perspective!

REVIEWER #3: This manuscript addresses an important issue and is timely. 

3.1 However, at times, it verges on advocacy rather than scientific reporting. The authors might want to re-read the discussion with this in mind.

Response: We have reread the discussion and are happy with its contents. There is some advocacy here, but nothing more than drawing out policy and societal implications. We believe this is very much part of our job as engaged social scientists. Furthermore, none of the critical points we make are new – they only support prior conclusions of other scholars critical of aspects of end child marriage movement. 

3.2 There is a missing word in line 248 "such, a pregnant 17-year-old is labelled a ‘teen’, but if married, and especially if she from a low." Please insert "is"after 'she."

Response: Corrected.

---

## [Decision Letter · Decision Letter 1]

17 Aug 2020

What does the American public know about child marriage?

PONE-D-20-17482R1

Dear Dr. Lawson,

We’re pleased to inform you that your manuscript has been judged scientifically suitable for publication and will be formally accepted for publication once it meets all outstanding technical requirements.

Kind regards,

Mellissa H Withers, PhD, MHS

Academic Editor

PLOS ONE

Additional Editor Comments (optional):

Reviewers' comments:

Reviewer's Responses to Questions

**Comments to the Author**

1. If the authors have adequately addressed your comments raised in a previous round of review and you feel that this manuscript is now acceptable for publication, you may indicate that here to bypass the “Comments to the Author” section, enter your conflict of interest statement in the “Confidential to Editor” section, and submit your "Accept" recommendation.

Reviewer #1: All comments have been addressed

Reviewer #2: All comments have been addressed

2. Is the manuscript technically sound, and do the data support the conclusions?

Reviewer #1: Yes

Reviewer #2: Yes

3. Has the statistical analysis been performed appropriately and rigorously? 

Reviewer #1: Yes

Reviewer #2: Yes

4. Have the authors made all data underlying the findings in their manuscript fully available?

Reviewer #1: Yes

Reviewer #2: Yes

5. Is the manuscript presented in an intelligible fashion and written in standard English?

Reviewer #1: Yes

Reviewer #2: Yes

6. Review Comments to the Author

Reviewer #1: (No Response)

Reviewer #2: The science is sound, so this should be published. Moreover, the authors have responded to my proposed revisions intended to bring greater clarity to the prose.

7. PLOS authors have the option to publish the peer review history of their article (what does this mean?). If published, this will include your full peer review and any attached files.

Reviewer #1: No

Reviewer #2: No

---

## [Editor Report · Acceptance letter]

26 Aug 2020

PONE-D-20-17482R1 

What does the American public know about child marriage? 

Dear Dr. Lawson:

I'm pleased to inform you that your manuscript has been deemed suitable for publication in PLOS ONE. Congratulations! Your manuscript is now with our production department. 

Kind regards, 

on behalf of

Dr. Mellissa H Withers 

Academic Editor

PLOS ONE